

# Equations of state in generalized hydrodynamics

**Dinh-Long Vu[1] and Takato Yoshimura[2]**

**1** Institut de Physique Théorique, CEA Saclay, Gif-Sur-Yvette, 91191, France
**2** Department of Mathematics, King's College London, Strand, London WC2R 2LS, U.K.

## Abstract

We, for the first time, report a first-principle proof of the equations of state used in the hydrodynamic theory for integrable systems, termed generalized hydrodynamics (GHD). The proof makes full use of the graph theoretic approach to Thermodynamic Bethe ansatz (TBA) that was proposed recently. This approach is purely combinatorial and relies only on common structures shared among Bethe solvable models, suggesting universal applicability of the method. To illustrate the idea of the proof, we focus on relativistic integrable quantum field theories with diagonal scatterings and without bound states such as strings.


## 1 Introduction

Extending the notion of statistical mechanics to describe states that are far from equilibrium has been one of the foremost challenges in theoretical physics. Although a unified *modus operandi* to deal with a genuinely out of equilibrium state is still out of reach, transport in

many-body systems can serve as a fertile testbed to study rich and sometimes counter-intuitive physics arising in non-equilibrium states. In particular, transport phenomena in one dimensional quantum systems have drawn a plethora of interest in recent years, due partially to spectacular advances in experiments that can now probe the dynamics of the quantum many-body systems in one dimension in a controlled manner [1–3]. From the theoretical point of view, transport in one dimension is somewhat special in that most of them are expected to be anomalous (non-diffusive) [4–6]. There is, however, a class of one dimensional quantum systems that exhibits a variety of transport types: integrable systems. It has been known that not only a seemingly likely case, ballistic transport [7–9], but other type of transports such as diffusive and super-diffusive transports can in fact occur in integrable systems [10–13]. In order to provide a coherent understanding in the transport phenomena in integrable systems, a hydrodynamic approach that can account for an excess amount of conserved charges, coined generalized hydrodynamics (GHD), was recently proposed [14, 15]. GHD was originally capable of describing only the dynamics at the Euler scale (leading contribution of the derivative expansion with respect to the space coordinates), but was later extended to capture the sub-leading (diffusive) effect [16]. GHD is not only able to describe an array of inhomogeneous dynamics [17–21, 23–26], but is also amenable to coping with external potentials [28] which allows us to efficiently simulate cold atom gases in a confining potential [29]. Moreover applicability of GHD to classical integrable systems is also numerically confirmed [27]. Remarkably, it can even yield some exact results including exact Drude weights at any temperature [19, 25, 30]. Despite of its far-reaching power to predict complicated dynamics, at the Euler scale, the picture GHD gives is intuitively rather clear: when systems are in local equilibrium, quasi-particles propagate ballistically with the effective velocity $v^{\text{eff}}(\theta; x, t)$. Therefore the functional form of the effective velocity, which can be regarded as the equations of state for GHD, determines the dynamics, and was first presented in [14, 15]. We note that a proof shown in [14] that exploited crossing symmetry only, is in fact incomplete as the proof assumed analyticity of some function in TBA, which is not necessarily true for some generalized Gibbs ensembles (GGEs). Thus a full proof of the equations of state is still missing in GHD. So far, the validity of GHD, which is equivalent to that of the effective velocity, has been numerically confirmed for spin chains such as the XXZ spin-$\frac{1}{2}$ chain [17–20, 24, 26] and the Fermi-Hubbard model [25], and it is believed that GHD correctly captures the long-wavelength dynamics of any Bethe solvable systems. Nonetheless, a down-to-earth proof of $v^{\text{eff}}(\theta)$ is still highly-desired to complete the program of GHD, and it is the purpose of this paper to report such a proof for relativistic integrable field theories with diagonal scatterings of one or more particle species.

Our strategy is essentially depending on form factor expansions by means of the LeClair-Mussardo formula [31]. The formula allows us to represent the expectation value of a local operator as an infinite series. This series is universal in the sense that the expectation values of two operators differ only in their connected form factors. The problem of evaluating the equations of state boils down to a direct comparison between the connected form factors of the charge and that of the current. Such comparison can be carried out by a well-known relation [32] between the connected form factors and the symmetric ones. Although this relation in its analytic form can be used to verify the current-average hypothesis for a few particles, which was presented in [14], it quickly becomes intractable. In order to provide a proof at full generality, we employ an equivalent formulation of graph expansion. The main idea is to apply the matrix-tree theorem to write Gaudin-like determinants and their minors as sums over trees. The latter are easy to control due to their simple combinatorial structure. Similar technique has been used in [33–35] and recently in [36, 37] to evaluate other observables such as partition function, correlation function and g-function.

The structure of the paper is the following: In section 2 we quickly summarize GHD and its equations of state. The proof for the current average is covered in section 3, which consists

of four subsections. In 3.1 we present basic facts about form factors, in particular the relation between connected and symmetric ones. In 3.2 we provide basic tools in graph theory. The proof for a theory with a single type of particle is completed in 3.3 and is extended in 3.4 to a theory with more than one type of particles. Section 4 closes our paper with perspectives.

Throughout this article, we shall focus on (1+1)-D relativistic integrable field theories with no bound states.

## 2   GHD and current average

As emphasized in the introduction, our motivation to obtain an explicit expression of the current average comes from GHD. Here we briefly recall how the equation of states in GHD is expressed in terms of quasi-particle basis.

GHD is a framework to study the dynamics of integrable systems at the Euler scale [14,15][1]. At such scale, generically, many-body systems are expected to be in a state where *local entropy maximization* is realized. In such a state, physics is dominated by macroscopic processes protected by conserved charges, and the state potentially carry a current. In practice, this scale can be accessed by taking a scaling limit of infinitely many degrees of freedom (i.e. the ratio between a typical microscopic scale $l_{\text{mic}}$, say the inter-particle length, and a typical macroscopic scale $l_{\text{mac}}$ becomes zero: $\epsilon = l_{\text{mic}}/l_{\text{mac}} \to 0$) while scaling the space-time simultaneously $(x, t) \to (\epsilon^{-1}x, \epsilon^{-1}t)$, which amounts to focusing on physics occurring at an emergent large scale called the *fluid cell*. Note that depending on the exponent $\alpha$ of the scaling of $x$, $\epsilon^{-\alpha}x$, a different scaling limit can be obtained (e.g. diffusive scaling for $\alpha = 1/2$ and super diffusive scaling for $1/2 < \alpha < 1$). The powerful assumption of local entropy maximization then provides us an efficient way to evaluate correlation functions at the Euler scale [38]. In particular, the expectation value of a given local operator $\mathcal{O}$ is computed by $\langle \mathcal{O}(x, t) \rangle_{\text{Eul}} = \text{Tr}(\rho(x, t)\mathcal{O})$ with $\rho(x, t) = \exp(-\sum_i \beta_i(x, t)Q_i)/Z(x, t)$, where $Q_i = \int \mathrm{d}x\, q_i(x, 0)$ are the conserved charges. This suggests that, at the Euler scale, in order to solve the macroscopic continuity equations $\partial_t \langle q_i(x, t) \rangle_{\text{Eul}} + \langle j_i(x, t) \rangle_{\text{Eul}} = 0$, one only has to know the equilibrium form of the averages of densities $\langle q_i \rangle_{\vec{\beta}}$ and currents $\langle j_i \rangle_{\vec{\beta}}$ as functions of Lagrange multipliers $\vec{\beta}$: the Euler scale averages are then simply $\langle q_i(x, t) \rangle_{\text{Eul}} = \langle q_i \rangle_{\vec{\beta}(x,t)}$ and $\langle j_i(x, t) \rangle_{\text{Eul}} = \langle j_i \rangle_{\vec{\beta}(x,t)}$.

In integrable systems, one-point functions in any generalized Gibbs ensemble (GGE) are conveniently represented in the quasi-particle basis. For instance, the density average of a conserved charge $Q_i$ reads [39]

$$\langle q_i \rangle = \sum_a \int \mathrm{d}\theta\, \rho_{\text{p},a}(\theta)h_{i,a}(\theta), \tag{1}$$

where $\theta$ is a quasi-momentum that parametrizes quasi-particles, and $a$ specifies each particle species. Here, $h_{i,a}(\theta)$ is the one-particle eigenvalue of $Q_i$: $Q_i|\theta\rangle_a = h_{i,a}(\theta)|\theta\rangle_a$, and $\rho_{\text{p},a}(\theta)$ is the density of particle [40] that can be computed within thermodynamic Bethe ansatz (TBA). Now, we need to know how $\langle j_i \rangle$ looks like in order to solve the macroscopic continuity equations. In [14,15], the exact expression of $\langle j_i \rangle$ was proposed that

$$\langle j_i \rangle = \sum_a \int \mathrm{d}\theta\, \rho_{\text{p},a}(\theta)v_a^{\text{eff}}(\theta)h_{i,a}(\theta), \tag{2}$$

where $v_a^{\text{eff}}$ is the velocity of excitation over an equilibrium state. It satisfies

$$v_a^{\text{eff}}(\theta) = v_a^{\text{gr}}(\theta) + \sum_b \int \mathrm{d}\theta' \frac{\varphi_{ab}(\theta - \theta')\rho_{\text{p},b}(\theta)}{p_b'(\theta)}(v_b^{\text{eff}}(\theta') - v_a^{\text{eff}}(\theta)), \tag{3}$$

---

[1]See [16] for the recent extension of GHD to account for diffusive corrections to GHD.

where $v_a^{\text{gr}}(\theta)$ is the group velocity, and $\varphi_{ab}(\theta)$ is the differential scattering kernel that is related to the S-matrix of a given model as $\varphi_{ab}(\theta) = -\text{i}\text{d}\log S_{ab}(\theta)/\text{d}\theta$. In [14], a proof for relativistic integrable quantum field theories with diagonal scatterings was provided using crossing symmetry. This proof, however, has a flaw in the sense that it implicitly assumes the analyticity of the source term $w(\theta) = \sum_i \beta_i h_i(\theta)$ that drives the Yang-Yang equation (see (19) for the definition), which is not necessarily guaranteed for some GGEs. For instance, in a nonequilibrium steady state generated by gluing two initially disconnected integrable systems at equilibrium, $w(\theta)$ actually has a jump as a function of $\theta$, hence nonanalytic [14]. We stress that our proof does not require the assumption of analyticity of $w(\theta)$, and therefore is applicable to arbitrary (local) GGEs. The current formula (2) has also been extended to the XXZ spin-$\frac{1}{2}$ chain where strings are present without proof but with numerical verifications [15]. The form of $v_a^{\text{eff}}(\theta)$ can be in fact considered as equations of state for GHD. Recall that equations of state are relations that relate the density averages $\langle q_i \rangle$ and the current averages $\langle j_i \rangle$: $\langle j_i \rangle = \mathcal{F}_i(\{\langle q_k \rangle\})$. Since it is precisely what $v_a^{\text{eff}}(\theta)$ is doing, making (2) different from (1) by its very appearance in (2), the functional form of the effective velocity determines the relation between the density and current averages. In the next section, we shall present the first-principle proof of (2). We note that the main idea of our proof, which is the form factor expansion, is same as the one presented in the appendix in [14]. The crucial difference is that, in our proof, we prove a statement (see (24) below) that is equivalent to the current formula (2) for any number of particles, while in [14], only the cases of a few numbers of particles were worked out. This generalization is made possible by making full use of the powerful techniques of graph theory in the same spirit as in [36]. This proof should serve as a first satisfactory proof of (2), but we expect that there is a yet another way of proving it, which is not needing any explicit use of relativistic / gallilean invariance.

Before embarking on the proof, let us conclude this section by introducing the dynamical equation of GHD. It immediately follows by plugging (1) and (2) into the macroscopic continuity equations. Using the completeness of the space of $Q_i$, it reads

$$\partial_t \rho_{\text{p},a}(\theta) + \partial_x(v_a^{\text{eff}}(\theta)\rho_{\text{p},a}(\theta)) = 0. \tag{4}$$

This type of equation for the spectral parameter $\theta$ has been found in several different contexts. For instance, the hydrodynamic equation of the hard-rod gas is known to have a same form to (4) with a similar effective velocity as (3) [41]. This is in fact readily derived by a simple kinetic argument presented in [14, 41], providing the underlying physical picture as to why the effective velocity has to be of the form (3). Note that the classical soliton gases of the KdV equation are also governed by an equation similar to (4) on the large scale [42, 43].

## 3 The proof

### 3.1 LeClair-Mussardo formula

Let us suppose the theory we consider has $N$ particle species $a_i$ with masses $m_i$ each of which differ, and it has no internal degrees of freedom. The Hilbert space of a generic (1+1)-D relativistic quantum field theory has natural bases: asymptotic *in* and *out* states $|\theta_1, \cdots, \theta_n\rangle_{a_1, \cdots, a_n}^{in,out}$ parameterized by rapidities $\theta$, which in turn diagonalize all the conserved charges if the model is integrable (to fix the basis, we use the *out* state, i.e. the ordering $\theta_1 < \cdots < \theta_n$). Another salient feature of the model, due to integrability, is that the dynamics is governed by the S-matrices that are factorizable into the product of two-body scattering matrix $S(\theta_i, \theta_j) = S(\theta_i - \theta_j)$. The *form factor* (with $n$ particles) of a local operator $\mathcal{O}$ of the

model is then defined by

$$F_{a_1,\cdots,a_n}(\theta_1,\cdots,\theta_n) = \langle \text{vac}|\mathcal{O}(0)|\theta_1,\cdots,\theta_n\rangle_{a_1,\cdots,a_n}, \tag{5}$$

which is expected to satisfy the following axioms [44, 45]:

1. Relativistic invariance

$$F_{a_1,\cdots,a_n}(\theta_1+\eta,\cdots,\theta_n+\eta) = e^{s\eta}F_{a_1,\cdots,a_n}(\theta_1,\cdots,\theta_n), \tag{6}$$

where $s$ is the spin of the operator $\mathcal{O}$.

2. Watson's equations

$$F_{a_1,\cdots,a_k,a_{k+1},\cdots,a_n}(\theta_1,\cdots,\theta_k,\theta_{k+1},\cdots,\theta_n) = S_{a_k,a_{k+1}}(\theta_k-\theta_{k+1})$$
$$\times F_{a_1,\cdots,a_{k+1},a_k,\cdots,a_n}(\theta_1,\cdots,\theta_{k+1},\theta_k,\cdots,\theta_n) \tag{7}$$

$$F_{a_1,\cdots,a_n}(\theta_1+2\pi i,\cdots,\theta_n) = F_{a_2,\cdots,a_n,a_1}(\theta_2,\cdots,\theta_n,\theta_1). \tag{8}$$

3. Kinematic poles

$$-i\operatorname*{Res}_{\theta\to\theta'} F_{a,b,a_1,\cdots,a_n}(\theta+\pi i,\theta',\theta_1,\cdots,\theta_n) = \left(1-\delta_{ab}\prod_{k=1}^{n}S_{a,a_k}(\theta-\theta_k)\right)F_{a_1,\cdots,a_n}(\theta_1,\cdots,\theta_n). \tag{9}$$

Note that if our model has bound states, the case which we do not consider here, there are additional *dynamical* poles on the imaginary axis within the strip $0 < \operatorname{Im}\theta_{ij} < \pi$. We further assume that form factors are meromorphic functions except poles that are dictated by the axioms mentioned. In what follows, for brevity, we shall focus on the case with one particle species.

The form factors generally serve as building blocks of more complicated matrix elements. Of particular interest for our purpose is the diagonal matrix element $\langle \overleftarrow{\theta} |\mathcal{O}(0)|\overrightarrow{\theta}\rangle$, where we introduced the shortened notation $|\overrightarrow{\theta}\rangle = |\theta_1,\cdots,\theta_n\rangle$ (and $\langle\overleftarrow{\theta}| = \langle\theta_1,\cdots,\theta_n|$). By use of the crossing relation, all form factors can be expressed in terms of the following form factor [32]

$$F_{2n}(\theta_1+\pi i+\delta_1,\theta_2+\pi i+\delta_2,\cdots,\theta_n+\pi i+\delta_n,\theta_n,\cdots,\theta_2,\theta_1)$$
$$= \prod_{i=1}^{n}\frac{1}{\delta_i}\sum_{i_1=1}^{n}\sum_{i_2=1}^{n}\cdots\sum_{i_n=1}^{n}f_{i_1,i_2,\cdots,i_n}(\theta_1,\cdots,\theta_n)\delta_{i_1}\delta_{i_2}\cdots\delta_{i_n} + \cdots, \tag{10}$$

where $f_{i_1,i_2,\cdots,i_n}(\theta_1,\cdots,\theta_n)$ is completely symmetric with respect to a rearrangement of indices, and "$\cdots$" refers to all the terms that vanish upon taking $\{\delta_i\}\to 0$ in any order. This form factor, in fact, is not well-defined due to the presence of kinematic singularities, i.e. its value depends on the order of limits $\{\delta_i\}\to 0$. There are two common ways to eliminate such singularities: one is to keep only finite terms by getting rid of all the terms that are divergent when taking $\{\delta_i\}\to 0$ in (10), yielding the *connected* form factor [31]

$$F_{2n}^{c}(\theta_1,\cdots,\theta_n) = \operatorname{FP}\lim_{\{\delta_k\}\to 0} F_{2n}(\theta_1+\pi i+\delta_1,\cdots,\theta_n+\pi i+\delta_n,\theta_n,\cdots,\theta_1). \tag{11}$$

Another one is to take a uniform limit such that $\delta_i = \delta \to 0$ for all $i$ that gives rise to the *symmetric* form factor [32]

$$F_{2n}^{s}(\theta_1,\cdots,\theta_n) = \lim_{\{\delta_k\}=\delta\to 0} F_{2n}(\theta_1+\pi i+\delta,\cdots,\theta_n+\pi i+\delta,\theta_n,\cdots,\theta_1). \tag{12}$$

These two form factors play essential roles in our proof later, and they are in fact related by the following relation

$$F_{2n}^{s}(\theta_1,\cdots,\theta_n) = \sum_{\substack{\alpha\subset\{1,\cdots,n\}\\ \alpha\neq\varnothing}} \mathcal{L}(\alpha|\alpha)F_{2|\alpha|}^{c}(\{\theta_i\}_{i\in\alpha}), \tag{13}$$

where $|\alpha|$ denotes the cardinal of the subset $\alpha$ and $\mathcal{L}(\alpha|\alpha)$ is the principal minor obtained by deleting the $\alpha$ rows and columns of the following matrix

$$L(\theta_1,\cdots,\theta_n)_{jk} = \delta_{jk}\sum_{l\neq j}\varphi_{j,l} - (1-\delta_{jk})\varphi_{j,k}, \tag{14}$$

where $\varphi_{i,j} = \varphi(\theta_i - \theta_j)$. Relation (13) can be obtained by considering two equivalent ways of writing the finite-volume diagonal matrix element $\langle\overleftarrow{\theta}|\mathcal{O}(0)|\overrightarrow{\theta}\rangle_V$ [32]

$$\sum_{\substack{\alpha\subset\{1,\cdots,n\}\\ \alpha\neq\varnothing}} F^{\mathrm{s}}_{2|\alpha|}(\{\theta_i\}_{i\in\alpha})\mathcal{G}(\{\theta_i\}_{i\in\bar\alpha}) = \sum_{\substack{\alpha\subset\{1,\cdots,n\}\\ \alpha\neq\varnothing}} F^{\mathrm{c}}_{2|\alpha|}(\{\theta_i\}_{i\in\alpha})\mathcal{G}(\alpha|\alpha), \tag{15}$$

where $\bar\alpha$ denotes the complementary of $\alpha$. On the left hand side, $\mathcal{G}(\theta_1,\cdots,\theta_n)$ is the determinant of the $n\times n$ Gaudin matrix

$$G(\theta_1,\cdots,\theta_n)_{jk} = \delta_{jk}\Big(Vp'(\theta_j) + \sum_{l\neq j}\varphi_{j,l}\Big) - (1-\delta_{jk})\varphi_{j,k}, \tag{16}$$

where $p'(\theta)$ is the derivative of the momentum $p$ with respect to the rapidity $\theta$. On the right hand side, $\mathcal{G}(\alpha|\alpha)$ is the principal minor of $G$ obtained by deleting its $\alpha$ rows and columns, and $V$ is the system size. Since the equality (15) is algebraic, it must hold for whatever value of $V$. Thus let us take a limit $V\to 0$. By writing the matrix $G(\{\theta_i\}_{i\in\bar\alpha})$ as the sum of a matrix of the type (14) and a diagonal matrix, we can write its determinent as a sum over partitions of $\bar\alpha$

$$\mathcal{G}(\{\theta_i\}_{i\in\bar\alpha}) = \sum_{\substack{I\subset\bar\alpha\\ I\neq\varnothing}}\mathcal{L}(I|I)\prod_{i\in I}Vp'(\theta_i). \tag{17}$$

Since $I$ is always a non-empty set, (17) necessarily vanishes when $V\to 0$ except when $\bar\alpha=\varnothing$, i.e. $\alpha=\{1,\cdots,n\}$. Hence the LHS of (15) becomes that of (13) under the limit. Together with an immediate observation that $\mathcal{G}(\alpha|\alpha)\to\mathcal{L}(\alpha|\alpha)$ with $V\to 0$, the relation (13) is established.

Having these in mind, we are now in a position to introduce the LeClair-Mussardo formula. The formula is a variant of spectral decomposition for the thermal (GGE) average of a local operator. It reads [31]

$$\langle\mathcal{O}\rangle = \frac{1}{Z}\mathrm{Tr}(e^{-\sum_i\beta_iQ_i}\mathcal{O}) = \sum_{l=0}^\infty\Big(\prod_{k=1}^l\int\frac{\mathrm{d}\theta_k}{2\pi}n(\theta_k)\Big)F^{\mathrm{c}}_{2l}(\mathcal{O};\theta_1,\cdots,\theta_k), \tag{18}$$

where $Z$ is the partition function, and the filling function $n(\theta) = 1/(1+e^{\varepsilon(\theta)})$ is given by the pseudo-energy $\varepsilon(\theta)$ that satisfies the Yang-Yang equation

$$\varepsilon(\theta) = \sum_i\beta_ih_i(\theta) - \int\frac{\mathrm{d}\theta'}{2\pi}\varphi(\theta-\theta')\log(1+e^{\varepsilon(\theta')}). \tag{19}$$

This is a remarkable simplification in computing the GGE average of a local operator, but it still requires us to compute the connected form factor, which is always a formidable task. Further, even if one manages to do so, carrying out the resummation is, in most cases, not feasible. Therefore, in practice, the formula is used with truncating after some terms; if excitation is small enough, this provides a fairly good approximation of the average.

Being said so, there are some known cases where one can evaluate the formula explicitly. One of such examples is the density of a conserved charge $Q = \int\mathrm{d}x\,q(x,0)$: the connected form factor of which is given by [39]

$$F^{\mathrm{c}}_{2n}(q;\theta_1,\cdots,\theta_n) = h(\theta_1)\varphi_{1,2}\cdots\varphi_{n-1,n}p'(\theta_n) + \mathrm{perm}, \tag{20}$$

where perm. is understood as permutations with respect to the integer set $\{1, \cdots, n\}$. Putting this into (18), we obtain an alternative expression of (1) [39]

$$\langle q \rangle = \int \frac{\mathrm{d}p(\theta)}{2\pi} n(\theta) h^{\mathrm{dr}}(\theta), \tag{21}$$

where the dressing operation is defined for any function $f(\theta)$ as

$$f^{\mathrm{dr}}(\theta) = f(\theta) + \int \frac{\mathrm{d}\theta'}{2\pi} \varphi(\theta - \theta') n(\theta') f^{\mathrm{dr}}(\theta'). \tag{22}$$

In the main proof, we will observe that, in fact, the same structure holds for the current operator $j$ as well. Recalling (2), we can also recast it into the similar form and expand as

$$\langle j \rangle = \int \frac{\mathrm{d}E(\theta)}{2\pi} n(\theta) h^{\mathrm{dr}}(\theta) = \sum_{l=0}^{\infty} \left( \prod_{k=1}^{l} \int \frac{\mathrm{d}\theta}{2\pi} n(\theta_k) \right) h(\theta_1) \varphi_{1,2} \cdots \varphi_{k-1,k} E'(\theta_k), \tag{23}$$

where $E'(\theta)$ denotes the derivative of the energy $E$ with respect to the rapidity $\theta$. This suggests that if the connected form factor of $j$ takes the following form, then (2) follows:

$$F_{2n}^{\mathrm{c}}(j; \theta_1, \cdots, \theta_n) = h(\theta_1) \varphi_{1,2} \cdots \varphi_{n-1,n} E'(\theta_n) + \mathrm{perm}, \tag{24}$$

which is the actual statement we are going to prove in order to establish (2).

### 3.2 Graphical representation

The relation (13) between the symmetric and connected form factors can be understood graphically. The matrix $L$ whose minors appear in this relation is a Laplacian matrix

$$L_{jk} = \delta_{jk} \sum_{l \neq j} \varphi_{j,l} - (1 - \delta_{jk}) \varphi_{j,k}. \tag{25}$$

It is the discretized Laplacian operator $\Delta$ on a graph in which a weight $\varphi_{j,k}$ is assigned to the edge connecting $j$ and $k$. Although $L$ has a vanishing determinant, as the elements on each row sum up to zero, its principal minors can be expressed as a sum over trees. This is the virtue of the matrix-tree theorem [46], also known as Kirchhoff theorem.[2]

**Definition 3.1** (Trees and forests). *A tree is a connected graph without cycles. A forest is a set of trees.*

In this paper we are referring to undirected graphs which means there is no direction on the edges. The reason behind is the symmetric property of the scattering differential: $\varphi_{i,j} = \varphi_{j,i}$. In the non-symmetric case, the definition of (directed) trees should be modified [36].

**Theorem 3.1** (Weighted matrix-tree theorem). *Let $\alpha$ be a subset of vertices $\{1, 2, \cdots, n\}$. Then we have*

$$\mathcal{L}(\alpha|\alpha) = \sum_{F \in \mathcal{F}_\alpha} \prod_{e \in F} \varphi_e, \tag{26}$$

*where the summation is performed over all forests of n vertices each tree of which contains exactly one vertex from $\alpha$. The product runs over all edges of the forests.*

---

[2]Matrix of the type (25) appeared in the work of Maxwell, one can therefore view the weights $\varphi_{jk}$ as the electric currents in a circuit.

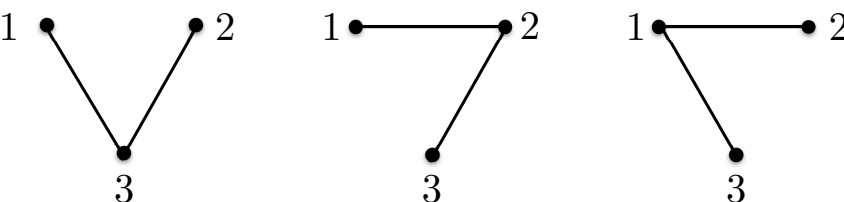

Figure 1: Trees associated with a minor of rank 2.

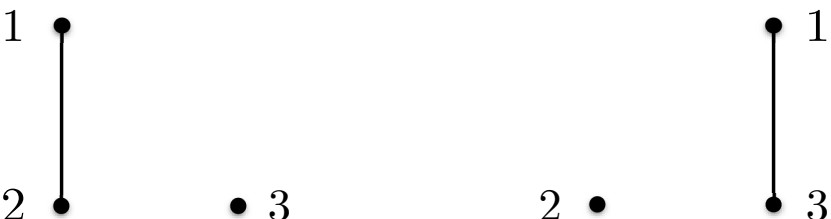

Figure 2: Forests associated with a minor $\mathcal{L}(\{2,3\}|\{2,3\})$.

This is known as the all-minor version of the matrix-tree theorem. A particular case is given by considering principal minors of rank $n-1$ i.e. by taking $\alpha$ to be one-element subsets. The forests would then become trees.

Let us illustrate the theorem in the case of three particles, where (25) is given by

$$L = \begin{pmatrix} \varphi_{1,2} + \varphi_{1,3} & -\varphi_{1,2} & -\varphi_{1,3} \\ -\varphi_{2,1} & \varphi_{2,1} + \varphi_{2,3} & -\varphi_{2,3} \\ -\varphi_{3,1} & -\varphi_{3,2} & \varphi_{3,1} + \varphi_{3,2} \end{pmatrix}. \tag{27}$$

All the principal minors of rank 2 are equal:

$$\mathcal{L}(1|1) = \mathcal{L}(2|2) = \mathcal{L}(3|3) = \varphi_{2,1}\varphi_{3,1} + \varphi_{2,1}\varphi_{3,2} + \varphi_{2,3}\varphi_{3,1}.$$

These terms are exactly the three trees spanning three vertices, see Fig.1. Note that we are referring to **labelled** trees. In particular, the trees in Fig.1 are considered as being distinguished, despite their similar combinatorial structure. The principal minors of rank 1 are written as forests with two trees. For example, when $\alpha = \{2,3\}$ we have $\mathcal{L}(\alpha|\alpha) = \varphi_{1,2} + \varphi_{1,3}$, as in Fig.2. The matrix-tree theorem provides a nice interpretation of the relation (13) between symmetric and connected form factors. For each subset $\alpha$ of $\{1,2,...,n\}$ we decorate the connected form factor $F^c_{2|\alpha|}(\{\theta_i\}_{i\in\alpha})$ by trees growing out of the elements of $\alpha$. The decorations must guarantee that all $n$ vertices are covered.

### 3.3 Main proof: one type of particle

Here we present a graph theoretic proof for (24). Our proof consists of three steps:

- Obtain the symmetric form factor of the charge $F^s_{2n}(q;\theta_1,\cdots,\theta_n)$ from the connected one (20) and the relation (13).

- Compute the symmetric form factor of the current $F^s_{2n}(j;\theta_1,\cdots,\theta_n)$ from that of the charge, by using the continuity equation.

- Find the connected form factor of the curent from the symmetric one, by going from the left hand side to the right hand side of equation (13).

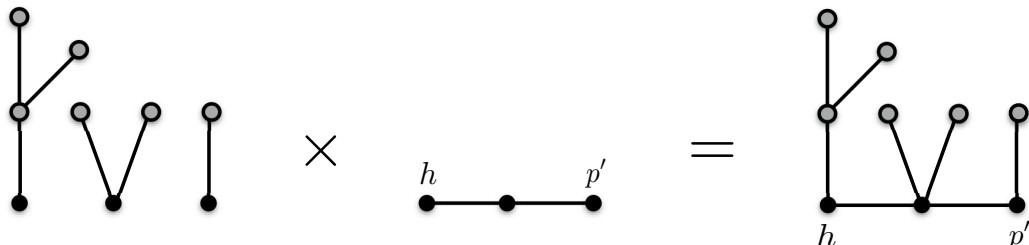

Figure 3: Pictorial representation of one of terms in the RHS of (13). Each term (forest) of $\mathcal{L}(\alpha|\alpha)$ and each term in $F^c_{2|\alpha|}(\{\theta_i\}_{i\in\alpha})$ form a spanning tree by merging together at vertices $\alpha$ represented by black dots.

The first and the last step are done with help of the matrix-tree theorem 3.1.

In the first step, we represent the connected form factor of the charge

$$F^c_{2n}(q;\theta_1,\cdots,\theta_n) = h(\theta_1)\varphi_{1,2}\cdots\varphi_{n-1,n}p'(\theta_n) + \text{perm}, \tag{28}$$

as $n!$ spines of length $n$ with the charge $h$ on one end and the momentum derivative $p'$ at the other end. Spines of length 1 with coinciding ends are allowed.

The corresponding symmetric form factor is obtained by decorating the spines with the trees, see Fig.3. Because the trees have different labelings and the spines come from different permutations, each term in the symmetric form factor is a (labelled) tree with two marked points, no tree appears more than once. Vice versa, each tree with two marked points can be decomposed to a spine and a forest. Indeed, the connectedness guarantees the existence of a path between the two marked points. Moreover, the uniqueness of this path is ensured by the non-existence of cycles. We conclude that the symmetric form factor of the charge is given by the sum over all the trees of $n$ vertices, with the weights $h$ and $p'$ inserted at two arbitrary vertices. This sum factorizes into the sum over the weights and the sum over the unmarked trees:

$$F^s_{2n}(q;\theta_1,\cdots,\theta_n) = \sum_{j=1}^{n} h(\theta_j)\sum_{k=1}^{n} p'(\theta_k)\sum_{T\in\mathcal{T}}\prod_{e\in T}\varphi_e. \tag{29}$$

Here, $\mathcal{T}$ denotes the set of the trees of $n$ vertices. The sum over these trees are exactly given by the principal minor of rank $n-1$ of the matrix (25). For instance, in the case of three particles:

$$F^s_6(q;\theta_1,\theta_2,\theta_3) = (h_1+h_2+h_3)(p'_1+p'_2+p'_3)(\varphi_{2,1}\varphi_{3,1} + \varphi_{2,1}\varphi_{3,2} + \varphi_{2,3}\varphi_{3,1}).$$

We now turn to the second step. In order to relate (29) to the symmetric form factor of the current $F^s_{2n}(j;\theta_1,\cdots,\theta_n)$, where $j$ satisfies the continuity equation $\partial_t q + \partial_x j = 0$, we note that there is a relation between $F^s_{2n}(q;\theta_1,\cdots,\theta_n)$ and $F^s_{2n}(j;\theta_1,\cdots,\theta_n)$ which is a simple consequence of the continuity equation:

$$F^s_{2n}(j;\theta_1,\cdots,\theta_n) = \frac{\sum_k E'(\theta_k)}{\sum_k P'(\theta_k)}F^s_{2n}(q;\theta_1,\cdots,\theta_n), \tag{30}$$

where we recall $E(\theta) = m\cosh\theta$ and $p(\theta) = m\sinh\theta$. To see this, we first observe

$$\langle\text{vac}|j(x,t)|\overrightarrow{\theta},\overleftarrow{\theta'}\rangle = e^{-im\sum_{k=1}^{n}\left[(\cosh\theta_k+\cosh\theta'_k)t-(\sinh\theta_k+\sinh\theta'_k)x\right]}\langle\text{vac}|j(0,0)|\overrightarrow{\theta},\overleftarrow{\theta'}\rangle \tag{31}$$

and thus,

$$\langle\text{vac}|\partial_x j(x,t)|\overrightarrow{\theta},\overleftarrow{\theta'}\rangle = im\sum_{k=1}^{n}(\sinh\theta_k + \sinh\theta'_k)\langle\text{vac}|j(x,t)|\overrightarrow{\theta},\overleftarrow{\theta'}\rangle. \tag{32}$$

Using this, it then follows that

$$
\begin{aligned}
F^{\mathrm{s}}_{2n}(j;\theta_1,\cdots,\theta_n) &= \lim_{\delta\to 0}\langle\mathrm{vac}|j(x,t)|\overrightarrow{\theta},\overleftarrow{\theta'}\rangle \\
&= \lim_{\delta\to 0}\frac{-i}{m\sum_k(\sinh\theta_k+\sinh\theta'_k)}\langle\mathrm{vac}|\partial_x j(x,t)|\overrightarrow{\theta},\overleftarrow{\theta'}\rangle \\
&= \lim_{\delta\to 0}\frac{i}{m\sum_k(\sinh\theta_k+\sinh\theta'_k)}\langle\mathrm{vac}|\partial_t q(x,t)|\overrightarrow{\theta},\overleftarrow{\theta'}\rangle \\
&= \lim_{\delta\to 0}\frac{\sum_k(\cosh\theta_k+\cosh\theta'_k)}{\sum_k(\sinh\theta_k+\sinh\theta'_k)}\langle\mathrm{vac}|q(x,t)|\overrightarrow{\theta},\overleftarrow{\theta'}\rangle \\
&= \frac{\sum_k E'(\theta_k)}{\sum_k p'(\theta_k)}F^{\mathrm{s}}_{2n}(q;\theta_1,\cdots,\theta_n),
\end{aligned}
\tag{33}
$$

where we used the continuity equation when passing from the second line to the third line, and noted

$$
\langle\mathrm{vac}|\partial_t q(x,t)|\overrightarrow{\theta},\overleftarrow{\theta'}\rangle = -im\sum_{k=1}^{n}(\cosh\theta_k+\cosh\theta'_k)\langle\mathrm{vac}|q(x,t)|\overrightarrow{\theta},\overleftarrow{\theta'}\rangle,
\tag{34}
$$

when moving from the third to the fourth line. Here, $\delta$ is defined as before in order to take the uniform limit $\theta'_j = \theta_j + \pi i + \delta$.

Now, applying this relation to (29), it immediately follows that

$$
F^{\mathrm{s}}_{2n}(j;\theta_1,\cdots,\theta_n) = \sum_{j=1} h(\theta_j)\sum_{k=1} E'(\theta_k)\sum_{T\in\mathcal{T}}\prod_{e\in T}\varphi_e,
\tag{35}
$$

which is nothing but the summation over all the trees of $n$ vertices, this time with $h$ and $E'$ inserted at two arbitrary points. By applying the same logic as in the first step, we can write this as a sum over spines and decorating trees

$$
F^{\mathrm{s}}_{2n}(j;\theta_1,\cdots,\theta_n) = \sum_{\substack{\alpha\subset\{1,\cdots,n\} \\ \alpha\neq\varnothing}}\mathcal{L}(\alpha|\alpha)F^{\mathrm{c}}_{2|\alpha|}(j;\{\theta_i\}_{i\in\alpha}),
\tag{36}
$$

where the spines now have $h$ and $E'$ on two ends

$$
F^{\mathrm{c}}_{2n}(j;\theta_1,\cdots,\theta_n) = h(\theta_1)\varphi_{1,2}\cdots\varphi_{n-1,n}E'(\theta_n) + \mathrm{perm}.
\tag{37}
$$

This is the desired formula for the current connected form factor. □

Our proof makes use of the matrix-tree theorem to express all the determinants and minors in the relation (13) between connected and symmetric form factors as sums over trees. We believe this is the natural language to understand this relation, as shown by the simplicity of the proof. One can of course argue that, because the matrix-tree theorem is two-fold, quantities which are expressed in terms of trees can be written as determinants of some matrices as well. As mentioned above, this is indeed true for the symmetric form factor of the charge or the current. For instance, (29) can be equivalently written as

$$
F^{\mathrm{s}}_{2n}(q;\theta_1,\cdots,\theta_n) = \mathcal{L}(1|1)\sum_{j=1}^{n} h(\theta_j)\sum_{k=1}^{n} p'(\theta_k),
\tag{38}
$$

where $\mathcal{L}(1|1)$ is the principal minor, obtained by deleting the first row and column of the $n\times n$ matrix (25). Interested readers are invited to derive (38), starting from (20) and (13) without using the matrix-tree theorem.

## 3.4 Main proof: more than one type of particle

Our method can be extended to a purely elastic scattering theory with $N$ types of particles $a = 1, 2, .., N$ of masses $m_a$. We illustrate it in the case of $T_2$ model [47, 48].

The underlying conformal field theory of this model is the minimal model $\mathcal{M}_{2,7}$ with central charge $c = -68/7$. It involves two nontrivial primary fields denoted by $\Phi_{1,2}$ and $\Phi_{1,3}$ with conformal dimensions

$$h_{1,2} = \bar{h}_{1,2} = -\frac{2}{7}; \quad h_{1,3} = \bar{h}_{1,3} = -\frac{3}{7}.$$

The theory can be quantized on a cylinder of circumference $V$ with the corresponding conformal Hamiltonian

$$H_0 = \frac{2\pi}{V}\left(L_0 + \bar{L}_0 - \frac{c}{12}\right).$$

The $T_2$ model is the perturbation of this theory by a positive parameter along the $\Phi_{1,3}$ direction

$$H = H_0 + \lambda \int_0^V dx\, \Phi_{1,3}(0, x). \tag{39}$$

More generally, $T_n$ models are perturbations of minimal models $\mathcal{M}_{2,2n+3}$ by the same operator. They can be realized as particular reductions of the sine-Gordon model [48].

The $T_2$ model is a massive integrable quantum field theory with the mass spectrum

$$\lambda = \kappa m_1^{2-2h_{1,3}}, \quad m_2 = 2m_1 \cos(\pi/5),$$

where $\kappa$ is a dimensionless constant. Its scattering information is encoded in the two-body scattering matrices

$$S_{11}(\theta) = \left\{\frac{2}{5}\right\}_\theta, \quad S_{12}(\theta) = \left\{\frac{1}{5}\right\}_\theta \left\{\frac{3}{5}\right\}_\theta, \quad S_{22}(\theta) = \left\{\frac{2}{5}\right\}_\theta^2 \left\{\frac{4}{5}\right\}_\theta, \tag{40}$$

where

$$\{x\}_\theta \equiv \frac{\sinh\theta + i\sin\pi x}{\sinh\theta - i\sin\pi x}\ .$$

Let us denote the two types of particle by $a = 1, 2$ with the corresponding energy and momentum $E_a$ and $p_a$. The thermodynamics of the model is driven by the TBA equations:

$$\epsilon_a(\theta) = \beta E_a(\theta) - \sum_b \int \frac{d\theta'}{2\pi} \varphi_{ab}(\theta - \theta')\log[1 + e^{\epsilon_b(\theta')}],$$

where $\varphi_{ab}$ is the logarithmic derivative of the scattering matrix: $\varphi_{ab}(\theta) = -i\partial_\theta \log S_{ab}(\theta)$. Unitarity again ensures that $\varphi$ is symmetric on its arguments:

$$S_{ab}(\theta)S_{ba}(-\theta) = 1 \Rightarrow \varphi_{ab}(\theta) = \varphi_{ba}(-\theta) \quad a, b \in \{1, 2\}. \tag{41}$$

The LM series for one point function of a local operator is a direct generalization of (18):

$$\langle \mathcal{O} \rangle = \sum_{l,m=0}^\infty \left(\prod_{j=1}^l \int \frac{d\theta_j}{2\pi} n_1(\theta_j) \prod_{k=1}^m \int \frac{d\vartheta_k}{2\pi} n_2(\vartheta_k)\right) F_{2l,2m}^c(\mathcal{O}; \vec{\theta}, \vec{\vartheta}). \tag{42}$$

In this expression, $n_1$ and $n_2$ are the filling functions of each type of particle: $n_a = 1/(1 + e^{\epsilon_a})$, $\overrightarrow{\theta}$ and $\overrightarrow{\vartheta}$ denote the two sets of rapidities: $\overrightarrow{\theta} = \{\theta_1, \cdots, \theta_l\}$, $\overrightarrow{\vartheta} = \{\vartheta_1, \cdots, \vartheta_m\}$. The connected form factors are defined in a similar way as before:

$$F^{\mathrm{c}}_{2l,2m}(\overrightarrow{\theta}, \overrightarrow{\vartheta}) = \mathrm{FP} \lim_{\{\delta_k\} \to 0} F_{l,m,m,l}(\theta_1 + \pi\mathrm{i} + \delta_1, \cdots, \theta_l + \pi\mathrm{i} + \delta_l,$$

$$\vartheta_1 + \pi\mathrm{i} + \delta_{l+1}, \cdots, \vartheta_m + \pi\mathrm{i} + \delta_{l+m}, \vartheta_m, \cdots, \vartheta_1, \theta_l, \cdots, \theta_1).$$

The symmetric form factor is obtained by taking the uniform limit: $\delta_k = \delta \to 0$ for all $k$. In particular, the relation between symmetric and connected form factor now becomes

$$F^{\mathrm{s}}_{2l,2m}(\overrightarrow{\theta}, \overrightarrow{\vartheta}) = \sum_{\substack{\alpha \subset \{1,\cdots,l\} \\ \beta \subset \{1,\cdots,m\}}} \mathcal{L}(\alpha, \beta | \beta, \alpha) F^{\mathrm{c}}_{2|\alpha|,2|\beta|}(\overrightarrow{\theta}_\alpha, \overrightarrow{\vartheta}_\beta), \tag{43}$$

where $\mathcal{L}(\alpha, \beta | \beta, \alpha)$ is the principal minor obtained by deleting the $\alpha$ rows and columns of the first diagonal block and $\beta$ rows and columns of the second diagonal block of the following matrix

$$L(\overrightarrow{\theta}, \overrightarrow{\vartheta}) = \begin{pmatrix} A & B \\ C & D \end{pmatrix}, \tag{44}$$

$$A_{ij} = \delta_{ij}\Big[\sum_{k \neq i}^{l} \varphi_{11}(\theta_i - \theta_k) + \sum_{k=1}^{m} \varphi_{12}(\theta_i - \vartheta_k)\Big] - (1 - \delta_{ij})\varphi_{11}(\theta_i - \theta_j), \quad 1 \leq i, j \leq l,$$

$$B_{ij} = -\varphi_{12}(\theta_i - \vartheta_j) \quad 1 \leq i \leq l, 1 \leq j \leq m,$$

$$C_{ij} = -\varphi_{21}(\vartheta_i - \theta_j) \quad 1 \leq i \leq m, 1 \leq j \leq l,$$

$$D_{ij} = \delta_{ij}\Big[\sum_{k=1}^{l} \varphi_{21}(\vartheta_i - \theta_k) + \sum_{k \neq i}^{m} \varphi_{22}(\vartheta_i - \vartheta_k)\Big] - (1 - \delta_{ij})\varphi_{22}(\vartheta_i - \vartheta_j), \quad 1 \leq i, j \leq m.$$

Despite its block structure, this matrix is still a Laplacian matrix as each of its rows sums up to zero. The matrix-tree theorem 3.1 is still valid, allowing us to write the principal minors $\mathcal{L}(\alpha, \beta | \beta, \alpha)$ as a sum over $(|\alpha| + |\beta|)$-forests of $l + m$ vertices. Each vertex now carries an index $a \in \{1, 2\}$ to indicate the type of particle it stands for. A branch connecting a particle of type $a$ and rapidity $\theta$ and another of type $b$ and rapidity $\vartheta$ carries a weight of $\varphi_{ab}(\theta - \vartheta)$. The symmetry of the scattering differential (41) indicates that the graphs are undirected.

We now turn our attention to the case of a conserved charge $Q$ which acts diagonally on the basis of multiparticle states:

$$\langle \overleftarrow{\vartheta}, \overleftarrow{\theta} | Q | \overrightarrow{\theta}, \overrightarrow{\vartheta} \rangle = \frac{1}{L}[h_1(\theta_1) + \cdots + h_1(\theta_l) + h_2(\vartheta_1) + \cdots + h_2(\vartheta_m)]\langle \overleftarrow{\vartheta}, \overleftarrow{\theta} | \overrightarrow{\theta}, \overrightarrow{\vartheta} \rangle. \tag{45}$$

The connected form factor for a state with $l$ particles of the first type and $m$ particles of the second type is given by the sum over $(l+m)!$ ways of distributing the particles on a spine with the charge $h$ on one end and $p'$ on the other end. Compared with the previous result (20), we now have to keep track of the particle type $a$, the scattering differential $\varphi_{ab}$ and the weight $h_a$ and $p'_a$ in each permutation. Explicitly we have:

$$F^{\mathrm{c}}_{2l,2m}(q; \theta_1, \cdots, \theta_l, \vartheta_1, \cdots, \vartheta_m)$$

$$= \sum_{\sigma \in S_{l+m}} p'_{\sigma_1}(\eta_{\sigma_1})\varphi_{\sigma_1\sigma_2}(\eta_{\sigma_1} - \eta_{\sigma_2})\ldots\varphi_{\sigma_{l+m-1}\sigma_{l+m}}(\eta_{\sigma_{l+m-1}} - \eta_{\sigma_{l+m}})h_{\sigma_{l+m}}(\eta_{\sigma_{l+m}}), \tag{46}$$

where the notations are to be understood as follows

$$p_i = \begin{cases} p_1 \\ p_2 \end{cases}, \quad h_i = \begin{cases} h_1 \\ h_2 \end{cases}, \quad \eta_i = \begin{cases} \theta_i & \text{if } 1 \le i \le l \\ \vartheta_{i-l} & \text{if } l < i \le l+m \end{cases},$$

$$\varphi_{ij} = \begin{cases} \varphi_{11} & \text{if } 1 \le i,j \le l \\ \varphi_{12} & \text{if } 1 \le i \le l, l < j \le l+m \\ \varphi_{21} & \text{if } l < i \le l+m, 1 \le j \le l \\ \varphi_{22} & \text{if } l < i,j \le l+m \end{cases}.$$

All the arguments of the previous section are still valid, in particular the symmetric form factors for the charge is given by

$$F^c_{2l,2m}(q;\theta_1,...,\theta_l,\vartheta_1,...,\vartheta_m) = \Big[\sum_{j=1}^{l} h_1(\theta_j) + \sum_{k=1}^{m} h_2(\vartheta_k)\Big]\Big[\sum_{j=1}^{l} p'_1(\theta_j) + \sum_{k=1}^{m} p'_2(\vartheta_k)\Big] \sum_{T \in \mathcal{T}} \prod_{e \in T} \varphi_e,$$

(47)

The last sum runs over all trees of $l+m$ vertices, $l$ of which is of type 1 and $m$ is of type 2. It is given by any principal minors of rank $l+m-1$ of the matrix (44). The connected form factor of the current is given by (46), with $p'$ replaced by $E'$.

## 4   Conclusions

In this article, we provided a graph theoretic proof of the equations of state used in GHD in the case of relativistic integrable quantum field theories without bound states. The proof applies to purely elastic scattering theories with one or multiple types of particles for which the corresponding LeClair-Mussardo formulae are known. Having the proofs for those cases, an obvious question would be if our approach can be applicable for theories where bound states and/or particles with internal degrees of freedom are present, such as the sine-Gordon model. This would be possible once we are able to extend the notion of connected form factor, or equivalently the LeClair-Mussardo formula for such theories. Such extension is still in development [49] and we leave it to future investigation.

We exemplified the graph theoretic idea using relativistic integrable quantum field theories, but it also works for the nonrelativistic case, such as the Lieb-Liniger model, through taking appropriate non-relativistic limits [50]. Extension of our method to spin chains seems to necessitate more investigation as much less is known about connected and symmetric form factors in spin chains [51, 52].

## Acknowledgements

The authors would like to thank Benjamin Doyon and Balázs Pozsgay for useful discussions. D-L.V. thanks Didina Serban and Ivan Kostov for suggesting the graph expansion idea. T.Y. acknowledges the support from Takenaka Scholarship Foundation and the ERC grant NuQFT.

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
