# Peer review of "Equations of state in generalized hydrodynamics"

_SciPost Physics, doi:SciPost Phys. 6, 023 (2019)_

## Round 2 · Referee Report · Anonymous · 2018-11-9

Strengths
1- First principle proof of the `equation of states' of the recently introduced generalized hydrodynamics theory of quantum integrable systems
2- Elegant approach based on combinatorics
3- The proof is relatively concise
Weaknesses
1- Calculation restricted to relativistic integrable field theories with diagonal scattering (excluding spin chains etc.)
Report
This paper provides a proof of the `equation of states' of the generalized hydrodynamic (GHD) theory of quantum integrable systems. GHD was introduced to capture the time evolution of locally-equibrated states in quantum integrable systems, while taking into account all conservation laws. A key equation in GHD gives the effective velocity of quasi-particle excitations over an equilibrium state given by the thermodynamic Bethe ansatz. This velocity appears in the expectation value of the currents in a given generalized equilibrium state. The proof relies on form factors and on the LeClair-Mussardo formula, and recent results on tree expansion approaches to TBA. The proof is limited to relativistic integrable field theories with diagonal scattering, but this is not unexpected given the approach. This is a very nice result, stated in a clear and elegant way. I recommend publication in SciPost as is, up to a minor comment that the authors might choose to address.
Requested changes
1- As the authors note after equation (4), the hydrodynamics of classical hard rod gases and of classical soliton gases has the same form as the GHD of quantum integrable systems. It would be good to mention that the formula for the velocity eq (3) is exactly what one would naturally write down using a semi-classical soliton gas picture. While this is not a proof, this provides a clear physical interpretation of eq 3 which is currently missing in the manuscript.
Author: Dinh-Long VU on 2019-01-11 [id 401]
(in reply to Report 1 on 2018-11-09)We are thankful to the referee for recognizing the overall quality of our paper. Concerning the comment about the similarity between the equations of state in GHD and those in classical hard rod gases and classical soliton gases, indeed it is better to point out that one can actually write down them by means of a simple kinetic reasoning developed in [21] and [41]. We provided a sentence in the main text mentioning this point.
Author: Dinh-Long VU on 2019-01-11 [id 400]
(in reply to Report 2 on 2018-12-18)Thank you for the constructive comments. Let us first comment on the novelty of our proof. We would like to stress that neither of the two proofs presented in the original GHD paper [14] is as complete as our proof. The first one in the paper, which is making use of crossing symmetry only, is in fact invalid in some cases. In that proof, analyticity of the source term $w(\theta)$ that drives the Yang-Yang equation (a key equation in TBA) is assumed, but this is not necessarily true for some (local) GGEs. For instance, in a non equilibrium steady state generated by gluing two initially disconnected thermal systems, $w(\theta)$ actually has a jump as a function of $\theta$, hence nonanalytic. Our proof, on the other hand, is applicable to any GGEs as it does not necessitate analyticity of the source term. Concerning the second proof provided in the appendix in the paper, it is in fact what our proof is based on. A sketchy proof of the connected form factor for currents presented there was restricted to a few numbers of particles, and we generalized it to an arbitrary number of particles by means of graph theory. In order to emphasize these points, we moved the sentence in the footnote to the main text with expanded explanations on the novelty of our proof. Concerning the requested changes, here are our modifications 1. TBA is not defined in the abstract. We added its full name “thermodynamic Bethe ansatz”. 2. Since Eq. (10) is the starting point of the proof, I recommend the authors to comment on why that is the relevant form factor. We added a brief explanation : “By use of the crossing relation, all form factors can be expressed in terms of the following form factor”. The detail of this statement can be found in the provided reference. 3. The authors should not assume that the readers have their background. For example, the model considered in section 3.4 is not introduced in a satisfactory way. We provided more information about the perturbed conformal field theory of the T2 model, including its operator content and its Hamiltonian formalism. We also present the particle spectrum as a function of the perturbing parameter and the scattering theory of this model.

---

## Round 2 · Referee Report · Anonymous · 2018-12-18

Strengths
1- Elegant proof for that expectation value of a current in a class of quantum integrable systems
Weaknesses
1- A proof already existed
2- The paper is not self contained
3- The paper is not written for a general audience
Report
This paper reports an elegant proof for that expectation value of the currents in a stationary state of a(n integrable) relativistic field theory with diagonal scattering. The work is set in the context of nonequilibrium dynamics of inhomogeneous integrable systems, in the framework of the so-called "generalized hydrodynamics". The proof sounds correct, but the authors did not really point out the novelty with respect to the existing proof, which, incidentally, was exhibited about two years ago by one of the authors himself (together with his collaborators).
I recommend this paper for publication, but, in order to avoid this to be considered a marginal paper, I request the authors to stress the importance of their proof even more.
Requested changes
1- TBA is not defined in the abstract.
2- Since Eq. (10) is the starting point of the proof, I recommend the authors to comment on why that is the relevant form factor.
3- The authors should not assume that the readers have their background. For example, the model considered in section 3.4 is not introduced in a satisfactory way.

---

## Round 3 · Referee Report · Anonymous · 2019-1-11

Report

The changes made by the authors are satisfactory, and I recommend publication in SciPost.

---

## Round 3 · Referee Report · Anonymous · 2019-1-31

Report

I recommend this paper for publication in the present form.

---

## Round 3 · List of Changes

- TBA in abstract specified
- remark on the originality of the proof added in introduction and in section 2
- The role of elementary form factor explained
- T2 model presented in more detail
- Similarity with classical hard rod gases explained

---

## Editorial Decision

published